# Gamma-Tocotrienol Protects the Intestine from Radiation Potentially by Accelerating Mesenchymal Immune Cell Recovery

**DOI:** 10.3390/antiox8030057

**Published:** 2019-03-06

**Authors:** Sarita Garg, Ratan Sadhukhan, Sudip Banerjee, Alena V. Savenka, Alexei G. Basnakian, Victoria McHargue, Junru Wang, Snehalata A. Pawar, Sanchita P. Ghosh, Jerry Ware, Martin Hauer-Jensen, Rupak Pathak

**Affiliations:** 1Division of Radiation Health, Department of Pharmaceutical Sciences, College of Pharmacy, University of Arkansas for Medical Sciences, Little Rock, AR 72205, USA; GargSarita@uams.edu (S.G.); RSadhukhan@uams.edu (R.S.); SBanerjee@uams.edu (S.B.); vymchargue@ualr.edu (V.M.); WangJunru@uams.edu (J.W.); SAPawar@uams.edu (S.A.P.); mhjensen@uams.edu (M.H.-J.); 2Department of Pharmacology and Toxicology, University of Arkansas for Medical Sciences, Little Rock, AR 72205, USA; SavenkaAlenaV@uams.edu (A.V.S.); BasnakianAlexeiG@uams.edu (A.G.B.); 3Central Arkansas Veterans Healthcare System, Little Rock, AR 72205, USA; 4Armed Forces Radiobiology Research Institute, USUHS, Bethesda, MD 20814, USA; sanchita.ghosh@usuhs.edu; 5Department of Physiology and Biophysics, College of Medicine, University of Arkansas for Medical Sciences, Little Rock, AR 72205, USA; JWare@uams.edu

**Keywords:** radiation, immune cells, intestine, organoid, vitamin E

## Abstract

Natural antioxidant gamma-tocotrienol (GT3), a vitamin E family member, provides intestinal radiation protection. We seek to understand whether this protection is mediated via mucosal epithelial stem cells or sub-mucosal mesenchymal immune cells. Vehicle- or GT3-treated male CD2F1 mice were exposed to total body irradiation (TBI). Cell death was determined by terminal deoxynucleotidyl transferase dUTP nick end labeling (TUNEL) assay. Villus height and crypt depth were measured with computer-assisted software in tissue sections. Functional activity was determined with an intestinal permeability assay. Immune cell recovery was measured with immunohistochemistry and Western blot, and the regeneration of intestinal crypts was assessed with ex vivo organoid culture. A single dose of GT3 (200 mg/kg body weight (bwt)) administered 24 h before TBI suppressed cell death, prevented a decrease in villus height, increased crypt depth, attenuated intestinal permeability, and upregulated occludin level in the intestine compared to the vehicle treated group. GT3 accelerated mesenchymal immune cell recovery after irradiation, but it did not promote ex vivo organoid formation and failed to enhance the expression of stem cell markers. Finally, GT3 significantly upregulated protein kinase B or AKT phosphorylation after TBI. Pretreatment with GT3 attenuates TBI-induced structural and functional damage to the intestine, potentially by facilitating intestinal immune cell recovery. Thus, GT3 could be used as an intestinal radioprotector.

## 1. Introduction

Ionizing radiation (IR)-induced gastrointestinal damage during radiotherapy or due to accidental overexposure can produce significant morbidity and mortality. IR inflicts adverse effects by impairing cellular function and signaling and/or by inducing the death of various populations of cells in the irradiated intestinal microenvironment. This microenvironment contains epithelial stem and progenitor cells and underlying non-epithelial mesenchymal cells, such as immune cells (neutrophils, lymphocytes, and macrophages), stromal cells (fibroblasts, myofibroblasts, and smooth-muscle cells), endothelial cells, and neuronal cells [1]. The impaired function or loss of these cells after irradiation collectively contributes to the development of intestinal toxicity. Importantly, the extent of intestinal damage depends on various factors, including dose, dose rate, and the quality of radiation [2]. In addition, the surface area of exposed tissue, individual radio-sensitivity, and the general health and genetic makeup of an individual determine the magnitude of gut injury [2].

Continually proliferating intestinal epithelial cells are highly sensitive to IR, and exposure to radiation induces apoptosis, characterized by DNA fragmentation [3]. The loss of epithelial cells causes villi to shorten, compromising the structural integrity of the intestine. In addition, damage to the epithelial layer disrupts barrier integrity and enhances intestinal permeability, allowing luminal bacteria to invade the damaged tissue. IR also alters the level of tight junction-related proteins, which are known to maintain barrier integrity [4]. Bacterial invasion due to the loss of barrier function eventually triggers inflammation and sepsis. 

Immune cells such as macrophages, neutrophils, and lymphocytes play a critical role in neutralizing invading luminal pathogens [5]. Upon breakdown of the mucosal barrier, intestinal tissue-resident macrophages are activated and liberate pro-inflammatory cytokines and chemokines that attract neutrophils and monocytes. Immune cells also release soluble regenerative mediators to restore barrier integrity after intestinal injury [6,7,8]. A study by Lindemans et al. showed that innate and adaptive lymphoid cells promote intestinal stem cell regeneration by releasing IL-22 [9]. Macrophage-derived Wingless/Integrated (WNT) ligands, which are highly hydrophobic cysteine rich secreted proteins, were shown to protect intestinal epithelial cells after exposure to radiation [7], and neutrophils have been shown to promote intestinal mucosal barrier function and tissue repair by producing amphiregulin, a member of the EGFR ligand family, in a mouse model of colitis [10]. These data clearly indicate that different types of immune cells are critical for restoring intestinal homeostasis after injury. However, exposure to total body irradiation (TBI) significantly decreases the levels of various immune cells in the damaged gut and in the blood [11]. When fewer immune cells are present after irradiation, they cannot effectively eliminate bacteria that invade from the gut lumen by way of the compromised mucosal epithelial barrier, and they likely produce fewer epithelial regenerative signals. This, in turn, triggers acute inflammation, which is considered one of the critical early effects of radiation [12]. Early symptoms can produce delayed fibrotic changes and tissue remodeling. Therefore, developing strategies to minimize the loss of immune cells after TBI will not only help remove invading microbes, but such strategies could restore barrier integrity by inducing epithelial regenerative signals.

The vitamin E family member gamma-tocotrienol (GT3) is a natural antioxidant and potent radioprotector [13]; it has been shown to suppress IR-induced hematopoietic, vascular, and intestinal structural damage after TBI [14,15,16]. Although the exact mechanisms of GT3-mediated intestinal radiation protection are not clear, previous studies demonstrated that GT3 pretreatment preferentially upregulates the expression of anti-apoptotic genes and downregulates pro-apoptotic factors in irradiated intestinal tissue [17]. The same study also showed that there was significantly less TBI-induced DNA fragmentation in crypt cells from GT3-treated animals relative to vehicle-treated ones [17]. In addition, GT3 also improved endothelial cell activity by inducing the thrombomodulin-activated protein C axis [18,19], which is known to play a critical role in limiting radiation-induced intestinal damage [20,21]. These findings suggest that GT3 has the ability to protect intestinal epithelial and non-epithelial cells following irradiation. However, the role of GT3 in modulating the recovery of intestinal immune cells and barrier function after TBI is not known. Here, we report that a single dose of GT3 administered 24 h before TBI suppresses cell death, accelerates the recovery of intestinal immune cells, and augments barrier function.

## 2. Materials and Methods

### 2.1. Experimental Model, GT3 Treatment, Irradiation, Tissue Procurement, and Euthanasia

All animal studies were carried out in strict accordance with the recommendations in the Guide for the Care and Use of Laboratory Animals of the National Institutes of Health. The animal protocol (AUP #3823) was approved by the Institutional Animal Care and Use Committee of the University of Arkansas for Medical Sciences (UAMS). Male CD2F1 mice were obtained from Charles River Laboratories (Houston, TX); mice between 8 and 12 weeks of age were used for experiments. They were housed in conventional cages in a pathogen-free environment with controlled humidity, temperature, and a 12–12 light–dark cycle with free access to drinking water and standard chow (Teklad, Madison, WI, USA).

GT3 was prepared freshly before administration with 5% Tween-80 (Fisher Scientific, Waltham, MA, USA) in saline. A single dose of 100 µL vehicle or GT3 (200 mg/kg body weight (bwt)) was administered subcutaneously 24 h before TBI.

Mice (not anesthetized) were exposed to a single TBI dose in a Shepherd Mark I 137Cs irradiator (model 25, J. L. Shepherd & Associates, San Fernando, CA, USA). During irradiation, the mice were placed in a custom-made, well-ventilated aluminum chamber with a Plexiglas lid (J.L. Shepherd & Associates). The chamber was divided into eight equal ‘‘pie slice’’ compartments by dividers made of T-6061 aluminum with a gold anodized coating. One mouse per compartment was placed to provide ample space to move. The chamber was placed on a turntable rotating at six revolutions per minute to assure uniform irradiation. The average dose rate was 1.01 Gy per min and was corrected for decay each day. A total dose in the range of 8 to 12 Gy was delivered depending on the endpoint assessed. For example, the mice were exposed to 8 Gy to measure apoptosis of epithelial cells and recovery of the immune cells, since these cells are highly sensitive to radiation, while we used relatively higher radiation dose (10 Gy and 12 Gy) for the intestinal permeability assay to induce enough damage in the intestine. The dose for the intestinal permeability assay was selected based on our previous experience. All radiation experiments were performed in the morning to minimize possible diurnal effects.

For tissue harvest, mice were anesthetized with 60 mg/kg sodium pentobarbital (Abbott Laboratories, Chicago, IL, USA) administered intraperitoneally. Samples of the intestine from irradiated and un-irradiated mice were procured and fixed in methanol–Carnoy’s solution for histological and immunohistochemical studies or snap-frozen in liquid nitrogen for molecular analysis. After tissue harvest, mice were euthanized by CO_2_ asphyxiation followed by cervical dislocation to eliminate any reasonable doubt of survival.

### 2.2. TUNEL Assay

TUNEL assays were performed as described previously [22]. Tissue samples were fixed, dehydrated, and embedded in paraffin. 4 µm thick tissue sections were cut using microtome, dewaxed, rehydrated in phosphate-buffered saline (PBS). TUNEL assay was performed using a In Situ Cell Death Detection Kit (Roche Diagnostics, Indianapolis, IN, USA) following the manufacturer’s instructions. Intestinal tissue sections were incubated with a reaction mixture of terminal deoxynucleotidyl transferase (TdT) and fluorescein (FITC)-labeled precursor in cacodylate-based buffer for 1 h at 37 °C, rinsed three times with 0.05% Tween-20 in PBS, and mounted under a ProLong^®^ Antifade medium containing 4′,6-diamidino-2-phenylindole dihydrochloride (DAPI) (Invitrogen, Carlsbad, CA, USA). TUNEL specificity was controlled by substituting the mixture of TdT and probe with the cacodylate buffer. The green spectrum (FITC) and blue spectrum (DAPI) were used to detect TUNEL-positive cells and nuclei, respectively. Images were captured with an Olympus IX-81 microscope (Olympus America Inc., Center Valley, PA, USA) equipped with a digital Hamamatsu ORCA-ER camera (Hamamatsu Photonics K.K., Hamamatsu City, Japan) under 63× magnification. Slidebook 6.2 software (SciTech Pty Ltd., Preston, Australia) was used for image capture. The results were presented as the frequency of the area of TUNEL-positive nuclear DNA in the total DAPI-positive area of nuclear DNA calculated for individual cells.

### 2.3. Assessment of Villus Height and Crypt Depth

Intestinal tissue sections stained with hematoxylin and eosin (H&E) were used to measure villus height and crypt depth using a computer-assisted image analysis platform (Image-Pro Premier, Rockville, MD, USA). Intestinal tissues fixed in Methyl–Carnoy’s fixed and embedded in paraffin were cut into 2–4 μm sections with a microtome. The slides with tissue sections were de-waxed by placing in an incubator overnight set at 60 °C, cooled down to room temperature, dipped into hematoxylin solution for 30 s, washed with deionized water, stained with 1% eosin solution, dehydrated with two changes in 95% and 100% alcohol for 30 s each, washed with xylene, and finally mounted with low viscosity Permount™ mounting media (Thermo Fisher Scientific, Grand Island, NY, USA), Mucosal villus height was measured from the tip to the base of each villus, and crypt depth was measured from the crypt base to the top opening. All measurements were done with a 10× objective lens, and a total of five areas were measured for each sample.

### 2.4. In Vivo Intestinal Permeability Assay

Intestinal permeability was assessed using an *in vivo* FITC-labeled dextran method after 4 days of total body exposure to 10 Gy or 12 Gy in mice pre-treated with or without GT3. After anesthetizing the mice by isoflurane inhalation, a midline laparotomy was performed, and the renal artery and vein were ligated bilaterally. A 10 cm segment of the small intestine, located 5 cm distal to the ligament of Treitz, was isolated and tied off. One hundred microliters of 4 kDa FITC-dextran (25 mg/mL in PBS) was injected into the isolated intestine using a 30 gauge needle, and the abdominal incision was closed. The renal artery and vein were ligated to prevent loss of FITC-dextran via urine. Blood samples were collected retro-orbitally 90 min after infusion of FITC-dextran into the intestinal lumen. Plasma sample was separated from whole blood by centrifugation (8000 rpm, 10 min, 4 °C) and the FITC-dextran content was measured with a fluorescence spectrophotometer (Synergy HT, Bio-Tek Instruments, Winooski, VT, USA) at an excitation wavelength of 480 nm and an emission wavelength of 520 nm. Standard curves were prepared to determine the concentration of FITC-dextran in plasma samples taking into account the dilution factor.

### 2.5. Immunoblotting

Protein lysates were prepared from intestinal tissue as described previously [23]. Primary antibodies to β-actin (Cell Signaling Technology; Danvers, MA), p-AKT (Cell Signaling Technology), CD2 (Santa Cruz Biotechnology; Dallas, TX), and occludin (Abcam; Cambridge, MA, USA) were used at a 1:1000 dilution for overnight incubation at 4 °C. Goat anti-rabbit IgG-HRP (Cell Signaling Technology, Danvers, MA, USA) and goat anti-mouse IgG-HRP (Santa Cruz Biotechnology, Dallas, TX, USA) secondary antibodies were used at a 1:5000 dilution for 2 h incubation at room temperature. Western blots were developed on autoradiography film (GeneMate, Kaysville, UT, USA) using chemiluminescent substrate (Thermo Fisher Scientific, Grand Island, NY, USA). Densitometry analyses were performed with ImageJ software available at NIH website, and β-actin was used as the loading control.

### 2.6. Intestinal Crypt Isolation and Organoid Culture in Matrigel

Small intestines were dissected and opened longitudinally, scraped gently to eliminate intestinal villi, and washed with cold PBS. The intestines were cut into 5 mm pieces and further washed three times with cold PBS to remove all visible debris. The tissue fragments were kept in 2.5 mM EDTA-PBS and incubated for 1 h on ice. After that, the tissue fragments were washed and re-suspended in cold PBS containing 5% fetal bovine serum (FBS). Then, fragments were gently mixed up and down with a 10 mL pipette and allowed to settle down for 5 min. The supernatant was enriched with crypts. The crypts were collected by passing through a 70 μm cell strainer (Fisher Scientific). Then, the fractions were centrifuged at 250 g to remove individual cells. Isolated crypts were counted and pelleted; 150–300 crypts were mixed with 40 µL Matrigel (Corning, Tewksbury, MA, USA) and dropped into the center of a well in a 24-well plate. After the Matrigel polymerized, 500 µL of crypt culture medium (Stemcell Technologies, Vancouver, BC, USA) was added into each well of the 24-well plate and kept in a CO_2_ incubator for 7 days.

### 2.7. Immunohistochemistry

Immunohistochemical staining was performed with standard techniques using an avidin–biotin complex, diaminobenzidine chromogen, and hematoxylin counterstaining. Appropriate positive and negative controls were included. Immunohistochemical staining for myeloperoxidase (MPO, for neutrophils) and macrophages was performed, as described previously [11], on proximal segments of the jejunum obtained at 0 h, 4 days, 7 days, and 21 days (with and without GT3). Tissue sections were incubated with primary rabbit anti-MPO (1:100, Dako, Glostrup, Denmark) or monoclonal rat anti-macrophage antibodies (RM0029-11H3, 1:100, Abcam) for 2 hours at room temperature. This was followed by a 30 min incubation with biotinylated secondary goat anti-rabbit IgG (MPO) and rabbit anti-rat IgG (macrophage) antibodies, with specificity for respective primary antibodies, at a 1:400 dilution (Vector laboratories, Burlingame, CA, USA). The slides were further incubated with the horseradish peroxidase (HRP) labelled avidin–biotin complex (Vector Laboratories) at a 1:100 dilution for 45 min. The HRP activity was measured with 0.5 mg/ml 3,3-diaminobenzidine tetrahydrochloride (DAB-HCl) solution (Sigma-Aldrich, St. Louis, MO, USA) and 0.003% H_2_O_2_ in Tris-buffered saline (TBS; Cell Signaling Technology) that enables developing the color. DAB solution was prepared immediately prior to use by dissolving 10 mg of DAB-HCl in 15 mL of TBS. Immunoreactivity was quantified using a computerized image analysis software, called Image-Pro Premier (Media Cybernetics; Rockville, MD, USA) as described elsewhere [24]. Cells positive for MPO and macrophages were identified after setting the color thresholding. The number of positive cells per 10 fields at 40× magnification was considered as a single value for statistical analysis.

### 2.8. RNA Extraction, cDNA Preparation, and Quantitative Reverse-Transcription PCR (qRT-PCR)

Total RNA was purified from frozen tissue with the RNeasy Plus Mini Kit (Qiagen, Valencia, CA, USA), according to the manufacturer’s instructions, after homogenizing the samples in TRIzol^®^ Reagent (Life Technologies, Grand Island, NY, USA). cDNA was synthesized with a cDNA reverse-transcription kit (Applied Biosystems, Foster City, CA, USA) after treating with RQ-DNase I (Promega, Madison, WI, USA). Predesigned Taqman assays (Applied Biosystems) for the following mouse genes were used: Lgr5, Mm00438890_m1; Msi1, Mm01203522_m1; Bmi1, Mm03053308_g1; and 18S rRNA, Hs99999901_s1. The mRNA levels were normalized to eukaryotic 18S rRNA and calculated relative to control mice with the standard ^ΔΔ^Ct method.

### 2.9. Statistical Analysis

Results are expressed as means ± the standard error of the mean (SEM). Data were analyzed with Prism software (version 4.0; GraphPad, San Diego, CA). Multiple means were compared by ANOVA and pairwise comparisons were analyzed with the Student’s t-test. A two-sided value of *p* < 0.05 was considered statistically significant. We followed the Guide for the Use of the International System of Units (SI) as recommend by National Institute of Standards and Technology (https://physics.nist.gov/cuu/pdf/sp811.pdf).

## 3. Results

### 3.1. GT3 Pretreatment Attenuated Intestinal Cell Death, Maintained Villus Height, and Enhanced Crypt Depth in Irradiated Mice

Radiation-induced intestinal cell death, characterized by DNA fragmentation, is an important contributor to acute radiation syndrome and determines survivability. We used the TUNEL assay to measure irreversible cell death in mice 24 h after they were irradiated in the presence or absence of GT3. A representative image of TUNEL-positive cells in the intestine is shown in Figure 1A. We observed no difference in the frequency of TUNEL-positive cells in vehicle- or GT3-treated groups before irradiation (Figure 1B). However, there were significantly more TUNEL-positive cells after irradiation (frequency of TUNEL positive cells in sham-irradiated vehicle, 0.04 ± 0.01 vs. irradiated vehicle, 0.72 ± 0.09; *p* = 0.0004) (Figure 1B). Interestingly, treating the mice with GT3 before irradiation suppressed intestinal cell death relative to vehicle treatment (frequency of TUNEL positive cells in irradiated GT3, 0.16 ± 0.04 vs. irradiated vehicle, 0.72 ± 0.09; *p* < 0.0001) (Figure 1B).

The loss of epithelial cells after irradiation shortens the villi, disrupting the structural integrity of the intestine. We found no difference in villus height between the sham-irradiated vehicle- and GT3-treated groups (Figure 1C). However, exposure to 8 Gy TBI on day 4 significantly decreased the height of villi in vehicle-treated animals (sham-irradiated vehicle, 248.87 ± 16.40 μm vs. irradiated vehicle, 174.84 ± 8.41 μm; *p* = 0.002) (Figure 1C). GT3 administration 24 h before 8 Gy TBI attenuated this decrease in villus height (irradiated vehicle, 174.84 ± 8.41 μm vs. irradiated GT3, 217.90 ± 12.28 μm; *p* = 0.01) (Figure 1C).

An increase in crypt depth is associated with the survival and a higher proliferation rate of stem cells after intestinal injury. Here, we measured crypt depth 4 days after 8 Gy TBI. In the sham-irradiated groups, we observed no difference in crypt depth after vehicle- or GT3-treatment (Figure 1D). However, in the irradiated animals, GT3-treatment significantly increased crypt depth compared to vehicle-treatment (GT3-treated group, 114.69 ± 3.83 μm vs. vehicle-treated group, 81.89 ± 3.23 μm; *p* = 0.0004) (Figure 1D).

### 3.2. GT3 Pretreatment Restored the Intestinal Barrier and Upregulated the Level of Occludin Protein in Irradiated Intestinal Tissue

Disrupting the structural integrity of the intestine impairs its barrier function, making it permeable to luminal contents. We measured intestinal barrier function with a permeability assay 4 days after exposure to 10 Gy or 12 Gy TBI; permeability was quantified as the optical density of FITC-labeled dextran in the blood following administration in the intestinal lumen (Figure 2A). The rationale for using a relatively higher dose of radiation for this study is to make sure the barrier integrity is lost. In vehicle-treated mice, 10 Gy TBI significantly increased intestinal permeability (sham-irradiated vehicle, 4266.70 ± 220.62 ng/mL vs. 10 Gy vehicle, 11,463.72 ± 2176.74 ng/mL; *p* = 0.01), as did 12 Gy TBI (sham-irradiated vehicle, 4266.70 ± 220.62 ng/mL vs. 12 Gy vehicle, 15,996.03 ± 2665.01 ng/mL; *p* = 0.006) (Figure 2A). However, pretreating the animals with GT3 significantly attenuated this increase in permeability compared to vehicle-treatment after both 10 Gy TBI (10 Gy vehicle, 11,463.72 ± 2176.74 ng/mL vs. 10 Gy GT3, 5188.08 ± 154.39 ng/mL; *p* = 0.009) and 12 Gy TBI (12 Gy vehicle, 15,996.03 ± 2665.01 ng/mL vs. 12 Gy GT3, 7285.45 ± 1291.09 ng/mL; *p* = 0.02) (Figure 2A).

Because radiation-induced changes in the level of tight junction-related proteins also play a critical role in intestinal permeability, we measured occludin protein level in the intestines of mice irradiated with or without GT3 pretreatment. Occludin level was measured at 4 h, 1 d, and 4 d after 8 Gy TBI. Compared to sham-irradiated animals (0 h), TBI significantly down-regulated occludin level at 4 h (*p* < 0.0001), 1 d (*p* = 0.0003), and 4 d (*p* = 0.0002) after vehicle treatment. However, compared to vehicle-treated irradiated groups, GT3 pretreatment significantly upregulated occludin level at 4 h (*p* = 0.01), 1 d (*p* = 0.001), and 4 d (*p* = 0.0003) (Figure 2B); at day 4, the level of occludin level reached that of the sham-irradiated group (Figure 2B). Figure 2C shows representative immunoblot data of occludin level at different time points in intestinal tissue before and after TBI and with or without GT3 pretreatment.

### 3.3. GT3 Pretreatment Rescued Populations of Intestinal Neutrophils, Macrophages, and Lymphocytes

TBI is known to deplete immune cells in the intestine. Neutrophil and macrophage levels in the intestine were measured at 4 d, 7 d, and 21 d by immunohistochemistry after exposure to 8 Gy TBI; lymphocyte levels were measured at 4 h, 1 d, and 4 d by immunoblot after exposure to 8 Gy TBI (Figure 3). Because lymphocytes are highly sensitive to radiation, we measured them at earlier time points. Compared to the sham-irradiated group, TBI significantly decreased the population of neutrophils at 4 d (*p* = 0.0002), 7 d (*p* = 0.002), and 21 d (*p* = 0.03) (Figure 3A–E). Importantly, pretreating the animals with GT3 significantly increased the neutrophil population at 4 d (*p* < 0.0001), 7 d (*p* = 0.005), and 21 d (*p* = 0.0009) compared to vehicle-treatment (Figure 3E).

Compared to the sham-irradiated group, TBI significantly decreased the population of macrophages at 4 d in the vehicle-treated group (frequency of sham-irradiated vehicle, 193.8 ± 38.16 cells vs. irradiated vehicle, 37.83 ± 6.99 cells; *p* = 0.001) (Figure 3F–J); however, the macrophage population recovered at days 7 and 21, resembling that of the sham-irradiated group (Figure 3J). Pretreating the animals with GT3 significantly increased the macrophage population in the intestine on day 4 compared to vehicle-treatment (frequency of irradiated vehicle, 37.83 ± 6.99 cells vs. irradiated GT3, 100.17 ± 14.96 cells; *p* = 0.003) (Figure 3J).

Next, we assayed lymphocytes by immunoblotting for CD2 (a lymphocyte marker) in the intestinal tissue of sham-irradiated and irradiated animals pretreated with or without GT3 (Figure 3K–L). We observed a significant decrease in lymphocytes in vehicle-treated animals after TBI relative to the sham-irradiated group at 4 h (*p* = 0.002), 1 d (*p* = 0.007), and 4 d (*p* = 0.0001) (Figure 3K). However, pretreating the animals with GT3 significantly increased the lymphocyte population compared to vehicle-treatment at 1 d (*p* = 0.01) and 4 d (*p* = 0.001) after TBI, but no difference in lymphocyte level was observed at 4 h (Figure 3K). Figure 3L shows a representative immunoblot of CD2 at different time intervals.

### 3.4. GT3 Pretreatment Failed to Promote Ex Vivo Intestinal Organoid Formation after TBI

Next, we investigated whether pretreating animals with GT3 before irradiation enhanced the ability of crypt cells to form intestinal organoids ex vivo, to determine whether GT3 can protect epithelial cells in absence of underlying mesenchymal cells. Figure 4A,B show a representative spheroid and an organoid formed form intestinal crypt cells. Although GT3 pretreatment significantly enhanced spheroid formation, there were significantly fewer organoids in the GT3-treated group than the vehicle-treated group (Figure 4C). Overall, there was no difference between the vehicle- and GT3-treated groups for the total count of organoids and spheroids (Figure 4C). In addition, GT3 pretreatment did not affect the expression of various stem cell markers, including Lgr5, Msi1, and Bmi1, in the intestinal tissue of sham-irradiated or irradiated mice (Appendix A).

### 3.5. GT3 Pretreatment Enhanced AKT Phosphorylation on Day 4 after Irradiation

Immune cells, particularly neutrophils and lymphocytes, are capable of activating the β-catenin pathway, which enhances epithelial cell proliferation through AKT phosphorylation. Thus, we assayed AKT phosphorylation on day 4 after TBI in the intestinal tissue of sham-irradiated or irradiated mice with or without GT3 pretreatment. In the vehicle-treated groups, TBI induced a significant decrease in AKT phosphorylation compared to sham-irradiation (Figure 5A). However, GT3 significantly upregulated AKT phosphorylation in irradiated animals compared to sham-irradiated ones (Figure 5A). Figure 5B shows a representative immunoblot for phosphorylated AKT in the intestinal tissue.

## 4. Discussion

TBI is used to treat hematopoietic malignancies and solid tumors. Further, TBI and chemotherapeutic drugs are routinely used to prepare patients for hematopoietic stem cell transplants [25,26,27,28,29]. In addition, astronauts are continuously exposed to chronic doses of TBI during their entire period of a space mission. Finally, the risk of exposure to TBI during radiological warfare is unavoidable. Under any of these circumstances, TBI can adversely affect various organs [30,31], especially the intestine [32], which is highly sensitive to IR.

The pathogenesis of IR-induced intestinal damage is an exceedingly complex process, involving the elimination, aberrant interaction, or functional dysregulation of various cell types present at the damaged site. IR disrupts cellular redox homeostasis, induces the generation of reactive oxygen species, and enhances oxidative stress, which may trigger apoptotic or non-apoptotic cell death, characterized by DNA fragmentation [17,33]. Fragmented DNA is the hallmark of irreversible cell death and can be detected by TUNEL assay. The loss of mucosal epithelial cells disrupts the structural integrity of the intestine, characterized by villus blunting and a decrease in mucosal length. We observed a statistically significant decrease in villus height on day 4 after 8 Gy TBI in vehicle-treated animals relative to un-irradiated controls, suggesting that the structural integrity of the intestine was compromised. Similar to our current findings, various groups have demonstrated that 8 Gy TBI adversely affects the structural integrity of the intestine [15,34,35]. However, we did not observe a decrease in villus height if the mice received a single dose of subcutaneous GT3 24 h before TBI. One explanation is that GT3 suppresses radiation-induced apoptosis. Suman et al. (2013) demonstrated that administration of GT3 (200 mg/kg bwt) significantly suppressed apoptotic cell death in the intestine relative to controls after 11 Gy TBI [17]. They also showed that GT3 pretreatment upregulated an array of anti-apoptotic genes and downregulated pro-apoptotic ones [17]. We found that treating animals with GT3 24 h before TBI significantly reduced intestinal cell death. Moreover, GT3 pretreatment increased the average crypt depth (crypts contain intestinal stem cells), suggesting that more stem cells could have survived and proliferated after TBI. Indeed, Liu et al. (2016) identified a strong positive correlation between crypt depth and stem cell survival after irradiation [36]. These findings clearly suggest that GT3 has the potential to suppress structural damage to the intestines after TBI.

The radiation-induced apoptotic death of epithelial cells reduces the mucosal surface area, disrupts barrier integrity, increases intestinal permeability, and facilitates the invasion of luminal microbes to the intestinal tissue. Epithelial cells, specifically Lgr5-positive, crypt-base columnar epithelial stem cells and their progenitors, are highly sensitive to IR and rapidly undergo apoptotic and mitotic death after exposure, reducing the integrity of the mucosal barrier. We observed a dose-dependent increase in intestinal permeability after exposure to 10 Gy or 12 Gy TBI. Importantly, pretreating animals with GT3 significantly decreased this radiation-induced intestinal permeability. Radiation-induced changes at the level of tight junction-related proteins play a critical role in enhancing intestinal permeability [4,37,38]. Occludin is a major constituent of tight junction-related proteins and is critical for maintaining the integrity of the intestinal epithelial barrier. Occludin-deficient mice exhibit increased colonic mucosal barrier dysfunction after ethanol challenge [39]. Moreover, genetic knockdown of occludin in human colon cancer cells (Caco-2) promote the disruption of tight junctions and barrier dysfunction after exposure to acetaldehyde [39]. All of these data indicate that suppressing occludin increases intestinal permeability. Our present study demonstrated a significant decrease in occludin level in the intestinal tissue, and this likely increased intestinal permeability after irradiation. A similar decrease in occludin level has also been noted in the intestines of non-human primates after radiation exposure [4]. Interestingly, pretreating our mice with GT3 significantly upregulated occludin level, relative to vehicle treatment, 4 days after TBI, suggesting that GT3 helps restore barrier integrity, potentially by upregulating occludin. All of these data clearly indicate that GT3 pretreatment plays a critical role in reinstating both the structural and functional integrity of the intestine after irradiation. However, we do not know how GT3 exerts its positive effects in the intestinal mucosa and on stem cells.

After irradiation, the crosstalk between epithelial stem cells and immune cells at the damaged site is critical for restoring intestinal homeostasis and modulating radio-sensitivity [7,9]. For example, depleting macrophages enhances radiation lethality in mice [40]. Moreover, macrophages play important roles in the repair and regeneration of damaged intestinal tissue [41], potentially by: (1) modulating inflammatory responses, (2) neutralizing invading microbes, and (3) releasing regenerative signals to activate intestinal stem cells [7]. IR not only eliminates intestinal stem cells, but significantly depletes immune cell populations at the damaged site. We found that TBI substantially reduces, in a time-dependent manner, the levels of neutrophils, lymphocytes, and macrophages in the intestinal stroma after exposure to 8 Gy TBI, and this is in agreement with findings by Garg et al. (2010) [11]. The greatest decrease in immune cells was observed on day 4 after TBI. This loss could be the result of apoptotic or mitotic death due to irreparable radiation-induced DNA damage. However, GT3 treatment enhanced the recovery of immune cells in the intestine after irradiation, potentially restoring epithelial homeostasis.

When the intestines are damaged, immune cells, specifically neutrophils and lymphocytes, promote the growth and differentiation of crypt epithelial cells via AKT phosphorylation, which in turn activates the regenerative β-catenin pathway [42,43]. Sumagin et al. (2016) showed that neutrophils activate the β-catenin pathway, through AKT phosphorylation, in mucosal epithelial cells to promote wound healing [42]. Using a co-culture model, Dahan et al. (2008) demonstrated that lymphocytes induce AKT phosphorylation in intestinal epithelial cells within 30 min [43]. Indeed, we observed a significant increase in AKT phosphorylation in intestinal samples from the GT3 group, relative to the vehicle-treated group, and this could be a major mechanism by which GT3 protects the intestines from radiation damage.

Similarly, Saha et al. (2016) showed that macrophage-derived Wnt ligands suppress radiation damage in epithelial cells in mice by activating the β-catenin pathway [7]. We found that GT3 accelerated the recovery of macrophages in the intestine after irradiation. This increase in the macrophage population could exert positive effects on intestinal epithelial cells by releasing regenerative signals. All of these data suggest that the GT3-mediated recovery of mesenchymal immune cells may play a critical role in protecting the intestine from radiation damage.

To further investigate the ability of GT3 to enhance stem cell proliferation in the absence of mesenchymal cells, crypts were cultured ex vivo in intestinal organoid culture medium. Interestingly, GT3 failed to promote intestinal organoid formation, suggesting that GT3 protects the intestine potentially by modulating mesenchymal immune cells, not by exerting positive effects directly on epithelial cells. Previous studies demonstrated that immune cells promote the proliferation of intestinal epithelial cells and recovery after injury by activating the β-catenin pathway via AKT phosphorylation [42,43]. Indeed, we found that GT3 significantly upregulated post-irradiation AKT-phosphorylation in the intestine. These data suggest that GT3 could alter AKT phosphorylation to protect the intestines from radiation damage.

## 5. Conclusions

In conclusion, we found that GT3 suppresses radiation-induced structural and functional damage in the intestine, potentially by upregulating occludin level and by facilitating the recovery of mesenchymal immune cells after TBI. Thus, GT3 has the potential to be a non-toxic medical countermeasure to protect the intestines from radiation damage.

## Figures and Tables

**Figure 1 antioxidants-08-00057-f001:**
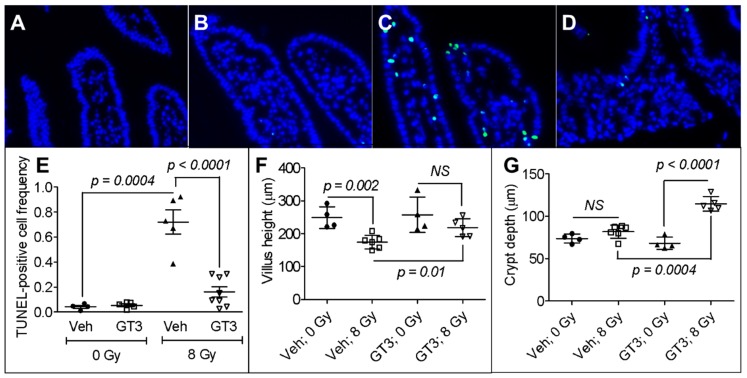
Effects of gamma-tocotrienol (GT3) pretreatment on total body irradiation (TBI)-induced intestinal damage. Representative photomicrograph of TUNEL-positive cells (green) in (**A**) vehicle (veh) un-irradiated, (**B**) GT3 un-irradiated, (**C**) vehicle irradiated, and (**D**) GT3 irradiated groups. (**E**) Frequency of TUNEL-positive cells in the intestine of sham-irradiated or irradiated mice, with or without GT3 pretreatment, TUNEL-positive cells were counted 24 h after 8 Gy TBI; (**F**) Villus height measured 4 d after 8 Gy TBI in sham-irradiated or irradiated groups, with or without GT3 pretreatment; (**G**) Crypt depth measured 4 d after 8 Gy TBI in sham-irradiated or irradiated groups, with or without GT3 pretreatment. Data represent the mean ± standard error of the mean (SEM). NS = not significant.

**Figure 2 antioxidants-08-00057-f002:**
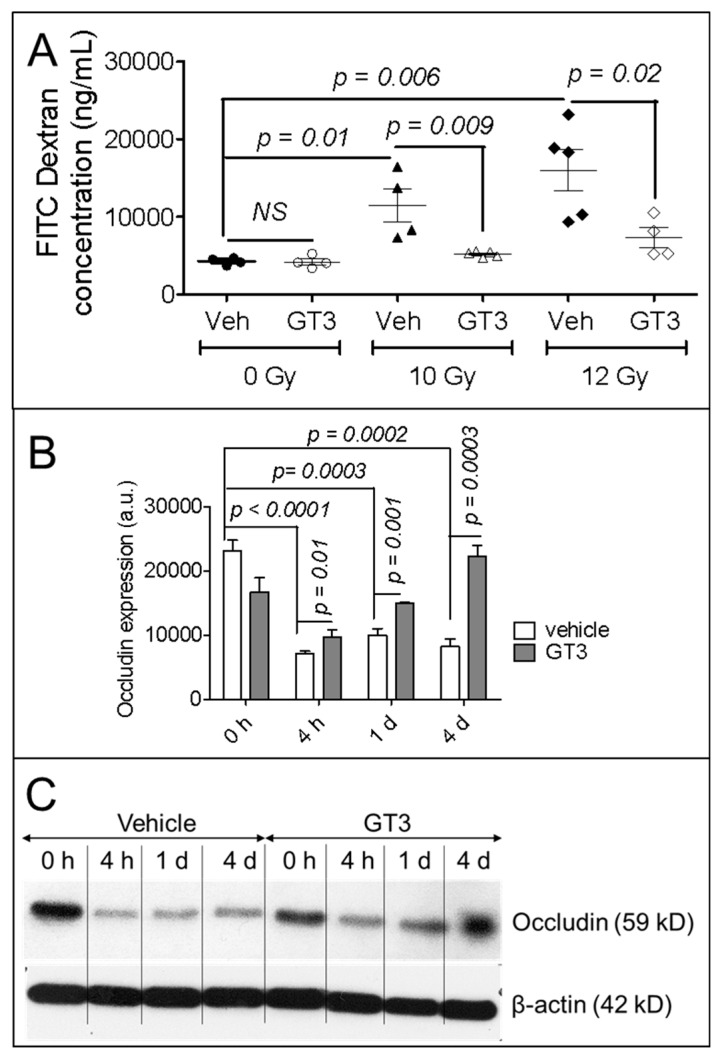
Effects of GT3 pretreatment on intestinal function. (**A**) Functional activity was measured on day 4 after 10 Gy or 12 Gy TBI with an intestinal permeability assay, as described in the Methods; (**B**) Densitometry of immunoblot showing time-dependent change in occludin in intestinal tissue from irradiated or sham-irradiated mice, with or without GT3 pretreatment 24 h before exposure to TBI; (**C**) Representative immunoblot of occludin at different time points. Data represent the mean ± SEM.

**Figure 3 antioxidants-08-00057-f003:**
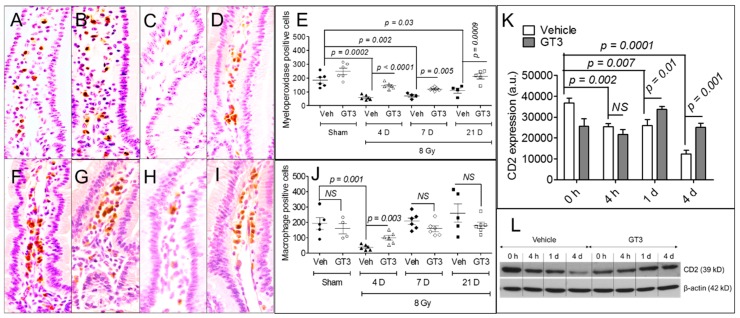
Effects of GT3 pretreatment on intestinal mesenchymal immune cells. (**A**–**D**) Representative photomicrographs (40× magnification) of myeloperoxidase immunostaining in samples from sham-irradiated vehicle, sham-irradiated GT3, irradiated vehicle, and irradiated GT3 groups on day 4 post-irradiation; (**E**) Myeloperoxidase-positive cells in immunostained tissue at different time points; (**F**–**I**) Representative photomicrographs (40× magnification) of macrophage immunostaining in samples from sham-irradiated vehicle, sham-irradiated GT3, irradiated vehicle, and irradiated GT3 groups on day 4 post-irradiation; (**J**) Cells with positive macrophage immunostaining at different time points; (**K**) Densitometry of immunoblot showing time-dependent change in CD2 in intestinal tissue of irradiated or sham-irradiated mice, with or without GT3 pretreatment 24 h before TBI; (**L**) Representative immunoblot of CD2 level at different time points. Data represent the mean ± SEM. NS = not significant.

**Figure 4 antioxidants-08-00057-f004:**
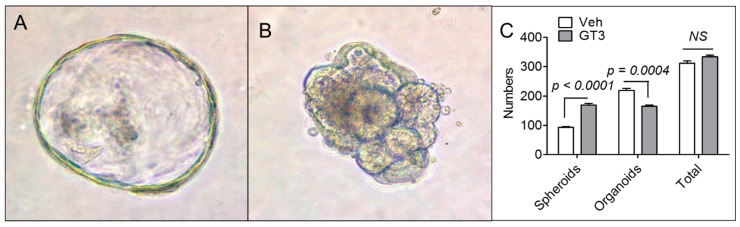
Effects of GT3 pretreatment on ex vivo intestinal organoid formation. (**A**) Representative photomicrograph of spheroid formed from intestinal crypt cells; (**B**) Representative photomicrograph of organoid formed from intestinal crypt cells; (**C**) Number of spheroids or organoids formed 7 days after exposure to 8 Gy TBI. Data represent the mean ± SEM. NS = not significant.

**Figure 5 antioxidants-08-00057-f005:**
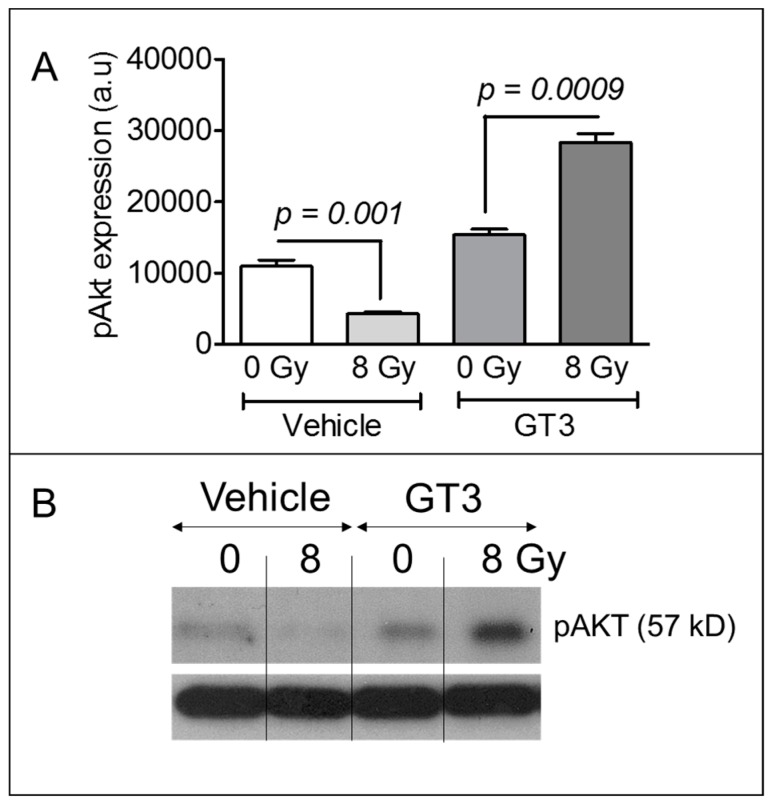
Effects of GT3 pretreatment on AKT phosphorylation in intestinal tissue. (**A**) Densitometry of an immunoblot showing change in AKT phosphorylation in intestinal tissue from irradiated or sham-irradiated mice, given either vehicle or GT3 24 h before exposure to TBI; (**B**) Representative immunoblot of phosphorylated AKT (pAKT) in intestinal tissue 4 d after 8 Gy TBI. Data represent the mean ± SEM.

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
