# Peer review of "Gamma-Tocotrienol Protects the Intestine from Radiation Potentially by Accelerating Mesenchymal Immune Cell Recovery"

_antioxidants, 2019, doi:10.3390/antiox8030057_

Round 1
Reviewer 1 Report
The purpose of this study was to evaluate the effects of gamma-tocotrienol to protects
intestinal epithelial and mesenchymal cells from intestine following irradiation. The authors measured apoptosis, villus height and crypt depth, intestinal permeability. Immune cells recovery was measured with immunohistochemistry and western blot.
They demonstrated that GT3 administered suppresses cell death, accelerates the recovery of intestinal immune cells, and augments barrier function.
-Page 3, line 105: The average dose rate was 1.01 Gy per min. Please indicate the total dose. Please explain the protocol of irradiation and justify because they use different doses of irradiation in each experiment.
-The authors use the TUNEL assay to measure cell death. However, they only represent the frequency of TUNEL-positive cells. Please, include the photomicrograph of all groups.
-In the figures 2, 3 and 5, they measure expression of occluding, CD2 and pAKT. However, the methodology of PCR to analyze expression is not explained in the text.
-The histopathology section should be more detailed including greater information on how the analysis was performed. The study of villus height and crypt depth would be performed by eosin-hematoxylin stain or electronic microscopy.
-Figure 2: include the photomicrograph of FITC dextran.
-Figure 2: please, include the densitometric quantification of western blot.
-Figure 3: please, include the densitometric quantification of western blot.
-Figure 5: please, include the densitometric quantification of western blot.
Author Response
Reviewer 1
Open Review
(x) I would not like to sign my review report
( ) I would like to sign my review report
English language and style
( ) Extensive editing of English language and style required
( ) Moderate English changes required
( ) English language and style are fine/minor spell check required
(x) I don't feel qualified to judge about the English language and style
Yes | Can be improved | Must be improved | Not applicable | |
Does the introduction provide sufficient background and include all relevant references? | (x) | ( ) | ( ) | ( ) |
Is the research design appropriate? | (x) | ( ) | ( ) | ( ) |
Are the methods adequately described? | ( ) | ( ) | (x) | ( ) |
Are the results clearly presented? | ( ) | ( ) | (x) | ( ) |
Are the conclusions supported by the results? | (x) | ( ) | ( ) | ( ) |
Comments and Suggestions for Authors
The purpose of this study was to evaluate the effects of gamma-tocotrienol to protects
intestinal epithelial and mesenchymal cells from intestine following irradiation. The authors measured apoptosis, villus height and crypt depth, intestinal permeability. Immune cells recovery was measured with immunohistochemistry and western blot.
They demonstrated that GT3 administered suppresses cell death, accelerates the recovery of intestinal immune cells, and augments barrier function.
Comment: Page 3, line 105: The average dose rate was 1.01 Gy per min. Please indicate the total dose. Please explain the protocol of irradiation and justify because they use different doses of irradiation in each experiment.
Response: We have mentioned the total dose and the rationale for selecting different doses of ionizing radiation to perform various endpoints in our revised version (Page 3, line 106 to 110).
Comment: The authors use the TUNEL assay to measure cell death. However, they only represent the frequency of TUNEL-positive cells. Please, include the photomicrograph of all groups.
Response: We agree with the reviewer and have incorporated representative photographs from each treatment group in Figure 1 of our revised version. We have also made the necessary corrections in the Figure 1 legend (Page 6).
Comment: In the figures 2, 3 and 5, they measure expression of occluding, CD2 and pAKT. However, the methodology of PCR to analyze expression is not explained in the text.
Response: For figure 2, 3, and 5, we have performed Western blot analysis and explained how the banding intensity was measured using Image J software in the Materials and Methods section (Page 4; line 168 & 169). PCR method was used to measure the expression of stem cell marker genes, which have been shown in Supplemental Figures.
Comment: The histopathology section should be more detailed including greater information on how the analysis was performed. The study of villus height and crypt depth would be performed by eosin-hematoxylin stain or electronic microscopy.
Response: We have incorporated a detail of histopathological staining method in our revised version as suggested by the reviewer (Page 3 & 4; line 139 to 144).
Comment: Figure 2: include the photomicrograph of FITC dextran.
Response: As described in the Materials and Methods section, we measured FITC dextran in the blood serum. We didn’t use any in vivo imaging system. But thanks to the reviewer for this suggestion. We will use this technique in our future experiments.
Comment: Figure 2: please, include the densitometric quantification of western blot.
Response: Densitometric quantification of western blot has been provided in 2B.
Comment: Figure 3: please, include the densitometric quantification of western blot.
Response: Densitometric quantification of western blot has been provided in 3K.
Comment: Figure 5: please, include the densitometric quantification of western blot.
Response: Densitometric quantification of western blot has been provided in 5A.

Reviewer 2 Report
Garg et al. analysed the effect of treatment of mice with gamma-tocotrienol before TBI.
The experiments showed clearly a positive effect of GT3 on the integrity of the villi,
However, it is not shown if the crypt stemm cells are really protected by this treatment. Immunohistochemical sections of the gut with KI67 staining or of cleaved Caspase-3. This staining would be very benificial to show the an increased survival of crypt cells after GT3 treatment and sould be included if tissues had been concerved.
It must be specially emphasised that the rules according the use of units (i. e. like https://physics.nist.gov/cuu/pdf/sp811.pdf) were applied ideally.
Author Response
Reviewer 2
x) I would not like to sign my review report
( ) I would like to sign my review report
English language and style
( ) Extensive editing of English language and style required
( ) Moderate English changes required
(x) English language and style are fine/minor spell check required
( ) I don't feel qualified to judge about the English language and style
Yes | Can be improved | Must be improved | Not applicable | |
Does the introduction provide sufficient background and include all relevant references? | (x) | ( ) | ( ) | ( ) |
Is the research design appropriate? | (x) | ( ) | ( ) | ( ) |
Are the methods adequately described? | (x) | ( ) | ( ) | ( ) |
Are the results clearly presented? | (x) | ( ) | ( ) | ( ) |
Are the conclusions supported by the results? | (x) | ( ) | ( ) | ( ) |
Comments and Suggestions for Authors
Comment: Garg et al. analysed the effect of treatment of mice with gamma-tocotrienol before TBI. The experiments showed clearly a positive effect of GT3 on the integrity of the villi.
Response: We would like to thank the reviewer for this comment.
Comment: However, it is not shown if the crypt stemm cells are really protected by this treatment. Immunohistochemical sections of the gut with KI67 staining or of cleaved Caspase-3. This staining would be very benificial to show the an increased survival of crypt cells after GT3 treatment and sould be included if tissues had been concerved.
Response: We agree with the reviewer on this comment. Actually we have already published the results of suggested experiment (Berbee et al. 2009), where our group had shown GT3 treatment enhances crypt cell survival compared to mice treated with vehicle. We exposed mice to 0, 8.5, 11, 13 and 15 Gy for crypt colony survival assay. The clear difference in crypt cell survival was noticed from 10 Gy onwards.
Comment: It must be specially emphasised that the rules according the use of units (i. e. like https://physics.nist.gov/cuu/pdf/sp811.pdf) were applied ideally.
Response: We would like to thank the reviewer for this comment. We have incorporated a sentence in our revised version as suggested by the reviewer (Page 5; line 212 to 214).
Citation
Berbée M, Fu Q, Boerma M, Wang J, Kumar KS, Hauer-Jensen M. Gamma-Tocotrienol ameliorates intestinal radiation injury and reduces vascular oxidative stress after total-body irradiation by an HMG-CoA reductase-dependent mechanism. Radiat Res. 2009 May; 171(5):596-605.

Round 2
Reviewer 1 Report
The authors respond that for figure 2, 3, and 5, they have performed Western blot analysis and explained how the banding intensity was measured using Image J software in the Materials and Methods section (Page 4; line 168 & 169). PCR method was used to measure the expression of stem cell marker genes, which have been shown in Supplemental Figures.
However, if the authors haven´t performed PCR and they only performed Western blot analysis they can´t say they measured occludin expression. With western blot you only can measure protein levels and not expression. Please correct this concept in all the manuscript
Author Response
The authors respond that for figure 2, 3, and 5, they have performed Western blot analysis and explained how the banding intensity was measured using Image J software in the Materials and Methods section (Page 4; line 168 & 169). PCR method was used to measure the expression of stem cell marker genes, which have been shown in Supplemental Figures.
However, if the authors haven´t performed PCR and they only performed Western blot analysis they can´t say they measured occludin expression. With western blot you only can measure protein levels and not expression. Please correct this concept in all the manuscript.
Response: We would like to thank the reviewer for giving the opportunity to correct the ambiguity. We have made the necessary corrections as suggested by the reviewer throughout the manuscript as indicated by highlight.
Reviewer 2 Report
The authors revised their manuscript appropriatly.
Author Response
The authors revised their manuscript appropriatly.
Response: We would like to thank the reviewer for this comment.